# Persistence of Anti-SARS-CoV-2 Antibodies Depends on the Analytical Kit: A Report for Up to 10 Months after Infection

**DOI:** 10.3390/microorganisms9030556

**Published:** 2021-03-08

**Authors:** Julien Favresse, Christine Eucher, Marc Elsen, Constant Gillot, Sandrine Van Eeckhoudt, Jean-Michel Dogné, Jonathan Douxfils

**Affiliations:** 1Department of Laboratory Medicine, Clinique St-Luc Bouge, 5004 Namur, Belgium; christine.eucher@slbo.be (C.E.); marc.elsen@slbo.be (M.E.); 2Department of Pharmacy, Namur Research Institute for Life Sciences, University of Namur, 5000 Namur, Belgium; constant.gillot@unamur.be (C.G.); jean-michel.dogne@unamur.be (J.-M.D.); jonathan.douxfils@unamur.be (J.D.); 3Department of Internal Medicine, Clinique St-Luc Bouge, 5004 Namur, Belgium; sandrine.vaneeckhoudt@slbo.be; 4Qualiblood s.a., 5000 Namur, Belgium

**Keywords:** COVID-19, SARS-CoV-2, antibody, kinetics, long-term, waning effect

## Abstract

Several studies have described the long-term kinetics of anti-SARS-CoV-2 antibodies but long-term follow-up data, i.e., >6 months, are still sparse. Additionally, the literature is inconsistent regarding the waning effect of the serological response. The aim of this study was to explore the temporal dynamic changes of the immune response after SARS-CoV-2 infection in hospitalized and non-hospitalized symptomatic patients over a period of 10 months. Six different analytical kits for SARS-CoV-2 antibody detection were used. Positivity rates, inter-assay agreement and kinetic models were determined. A high inter-individual and an inter-methodology variability was observed. Assays targeting total antibodies presented higher positivity rates and reached the highest positivity rates sooner compared with assays directed against IgG. The inter-assay agreement was also higher between these assays. The stratification by disease severity showed a much-elevated serological response in hospitalized versus non-hospitalized patients in all assays. In this 10-month follow-up study, serological assays showed a clinically significant difference to detect past SARS-CoV-2 infection with total antibody assays presenting the highest positivity rates. The waning effect reported in several studies should be interpreted with caution because it could depend on the assay considered.

## 1. Introduction

Currently, the revelation of SARS-CoV-2 RNA through a real-time reverse transcription polymerase chain reaction (RT-PCR) from nasopharyngeal swab samples is considered to be the gold standard method for the diagnosis of acute SARS-CoV-2 infection. Less invasive salivary samples have also been reported as an alternative to nasopharyngeal swab samples [1]. The targeted genes for RT-PCR detection may include a combination of *N*, *E*, *RdRp*, *orf1a* and *orf1b* genes.

The detection of anti-SARS-CoV-2 antibodies serves as an adjunct to molecular testing for the diagnosis of COVID-19 especially in patients who present late with a low viral load. Serological testing has been successfully used to evaluate seroprevalence, to identify convalescent plasma donors, to monitor herd immunity and for risk predictions [2,3,4] Antibody assessment and monitoring are also likely to play a key role in the context of the global vaccination strategy [5].

Compared with commercial immunoassays, only neutralization activity assays reliably measure the actual protective immunity of antibodies [6]. However, neutralization activity assays are only reserved for specialized laboratories and require a high workload, skillful operators, expensive installations, crucial biosafety measures and have, to date, a low throughput. Therefore, the use of fully automated immunoassays that have a well-demonstrated correlation with neutralization activity should be considered in the routine clinical setting [5].

Current serological assays use different SARS-CoV-2 antigen targets (i.e., nucleocapsid protein (NCP), spike proteins (S) and the receptor-binding domain (RBD)) and formats (i.e., immunoglobulin G (IgG), IgA, IgM or total antibodies). Most assays possess a unique target but multiplex assays have also been developed and validated [7,8]. The NCP participates in RNA packages and the release of virus particles while the transmembrane spike glycoprotein comprises two functional subunits responsible for binding to the host cell receptor (N-terminal S1 subunit) and for the fusion of the viral and cellular membranes (C-terminal S2 subunit) [9]. The RBD is located at the C-terminal region of the S1 subunit [10]. The RBD interacts with human cells that express angiotensin-converting enzyme 2 (ACE2) and induces the entry of the virus.

The antibody response to SARS-CoV-2 infection has been shown to be directed against multiple antigens of the virus including different epitopes of the spike protein. Antibodies targeting the RBD in the C-terminal region of the S1 subunit have been considered to be neutralizing [11]. Other antibodies target the NCP or non-structural proteins and their detection can be used as markers of recent infection [7,8].

Reports evaluating antibody persistence inconsistently mention a waning effect of the serological response [12,13,14]. Based on these observations, a few authors have claimed that cross-sectional seroprevalence studies to evaluate population immunity may underestimate rates of prior infections [12]. A recent report also suggests that changing the vaccine policy to give to previously infected individuals only one dose of the vaccine would not negatively impact their antibody response and may consequently free up many vaccine doses [15]. Therefore, the divergent opinions regarding antibody persistence warrant further investigations to ensure an accurate and reliable evaluation of the serological status of each individual because, depending on the vaccinal strategy that will be applied in the coming weeks, this could represent the saving of up to 100 million vaccine doses worldwide [16].

The aim of this study was therefore to explore the temporal dynamic changes of immune response after SARS-CoV-2 infection in hospitalized and non-hospitalized symptomatic patients for a period of up to 10 months using different analytical kits for SARS-CoV-2 antibody detection. This will permit the investigation of, and provide more insight into, the understanding of this possible waning effect.

## 2. Materials and Methods

The study protocol was in accordance with the Declaration of Helsinki and was approved by the Medical Ethical Committee of Saint-Luc (Bouge, Belgium; approval number B0392020000005).

### 2.1. Patients and Samples Collection

This study was conducted at the clinical biology laboratory of the Clinique Saint-Luc (Bouge, Namur, Belgium). A total of 201 samples from 84 patients with a confirmed SARS-CoV-2 RT-PCR were retrospectively included from 26 March 2020 to 6 January 2021. Information on the days since the onset of symptoms was collected from medical records. When data about symptoms were not available (n = 15), the day of diagnosis (i.e., RT-PCR result) was used instead. Different time intervals were also created to calculate the rate of positive samples (i.e., 0–15, 15–40, 41–100, 101–150, 151–200, 201–235 and 236–300 days). Blood samples were collected into serum-gel tubes (BD SST II Advance^®^, Becton Dickinson, NJ, USA) according to the standardized operating procedure and manufacturer recommendations. Samples were centrifuged for 10 min at 1740× *g* on a Sigma 3-16KL centrifuge. Sera were stored in the laboratory serum biobank at −20 °C from the collection date. Frozen samples were thawed for 1 h at room temperature on the day of the analysis. Re-thawed samples were vortexed before the analysis.

### 2.2. Analytical Procedures

Six commercial immunoassays were used to evaluate the long-term kinetics of antibodies. The characteristics of these assays are presented in Table 1. Each patient’s sample was analyzed on the six different assays. The results rendered below the limit of quantification (LOQ) of the assay were rounded to the LOQ of each assay to allow quantitative calculations and data processing.

The RT-PCR for SARS-CoV-2 determination in respiratory samples (nasopharyngeal swab samples) was performed on the LightCycler^®^ 480 Instrument II (Roche Diagnostics^®^) using the LightMix^®^ Modular SARS-CoV *E*-gene set.

### 2.3. Statistical Analyses

Descriptive statistics were used to analyze the data. A Mann–Whitney test was used to compare the different groups. The positivity rates were calculated as the proportion of SARS-CoV-2 positive samples by serological tests initially categorized as positive by the RT-PCR. A non-linear regression model with log-transformed data was used to compute the antibody kinetics since symptom onset (or diagnosis) using the following equation:$$\left[\frac{AUC\;*\;disappearance\;rate\;*\;appearance\;rate}{appearance\;rate\;-\;disappearance\;rate}\right]\;*\;[Exp\;\left(-\;disappearance\;rate\;*\;days\;since\;symptom\;onset\right)\;-\;Exp\;(-\;appearance\;rate\;\;*\;days\;since\;symptom\;onset)]$$

A survival analysis was also performed to estimate the cumulative probability of positive samples since symptom onset (or diagnosis) using a log-rank Mantel–Cox test comparison. Mantel-Haenszel hazard ratios were computed for between test comparison. Inter-rate agreements, i.e., agreement and Cohen’s kappa, and correlation studies were also determined. A *p* value < 0.05 was used as a significance level. Data analysis was performed using GraphPad Prism^®^ (version 9.0.1, California, CA, USA), MedCalc^®^ (version 14.8.1, Ostend, Belgium) and JMP^®^ software (version 15.2.1, Cary, NC, USA).

## 3. Results

### 3.1. Population Characteristics

Among the 84 individuals, 44 were females (median age = 46 years; min–max: 24–95 years) and 40 were males (median age = 61 years; min–max: 24–88 years). Multiple sequential sera were available for 55 patients and 17 required hospitalization (i.e., categorized as severe patients). Hospitalized patients were elder (median age = 74 years) compared with non-hospitalized patients (median age = 46 years; *p* value = 0.0007). The median time between the symptom onset and the RT-PCR was three days (interquartile range (IQR): 1–8 days).

### 3.2. Kinetics of Positivity Rates

In samples collected early since symptom onset (i.e., <15 days), positivity rates were low. The Roche NCP total antibody assay had the highest positivity rate in this time period (i.e., 69.2%) while the DiaSorin S1 + S2 IgG and the Phadia S1 IgG assays showed the lowest positivity rates (i.e., 38.5%). At the second time point, the highest positivity rates were observed for the Roche NCP and the Ortho S1 total assays (i.e., 96.3% and 100%, respectively). A gradual increase in positivity rates toward a plateau was observed for both the Roche RBD and the Ortho S1 total assays with the latter reaching the plateau earlier than the Roche RBD total antibody assay. The highest positivity rates for the other assays were observed at the fourth time point (i.e., for the Phadia S1 IgG and the Ortho S1 IgG assays) and at the fifth time point (i.e., for the Roche NCP total antibody and the DiaSorin S1 + S2 IgG assays) (Table 2). Overall, the total assays presented higher positivity rates and reached their highest positivity rates sooner compared with IgG assays. 

### 3.3. Kinetic Models of Serological Response

Figure 1 represents the level of antibody response by days after symptom onset according to severity. Depending on the assay and/or the population considered, a rapid increase in antibody titers was observed followed by a plateau phase or a decrease phase. After 300 days, hospitalized patients had overall a higher maximal response peak and a more persistent antibody response (e.g., the Roche NCP total antibody and the DiaSorin S1 + S2 IgG assays).

The C_max_ was consistently higher in hospitalized than in non-hospitalized patients (Table 3).

Figure 2 represents the cumulative probability of positive samples after 14 days until the last follow-up point, i.e., 300 days. Of this selected cohort of 188 samples collected after the fourteenth day, 2, 8, 10, 21, 25 and 57 samples were reported as negative for the Ortho S1 total antibody, the Roche RBD total antibody, the Roche NCP total antibody, the Ortho S1 IgG, the DiaSorin S1 + S2 IgG and the Phadia S1 IgG, respectively. Assays targeting total antibodies, i.e., Ortho S1 total antibody, Roche RBD total antibody and Roche NCP total antibody, had the highest cumulative probability of positive samples at the latest follow-up period compared with IgG assays (*p* value < 0.0001). The Ortho S1 total antibody assay was the only test that did not statistically differ from the 100% of cumulative positive probability, i.e., “all positive” on the figure at the latest follow-up point (*p* value = 0.1573). The Phadia S1 IgG was the only test having a cumulative probability of positive samples below 50%. The median survival was 239 days with this test.

A comparison of the Mantel–Haenszel hazard ratios between the different tests is reported in Table 4. Between the different assays, the Roche RBD total antibody and the Ortho S1 total antibody showed the highest agreement, Cohen’s kappa index and correlation coefficient (Table 4). An agreement of at least 95% was only reached for total assays.

## 4. Discussion

In this study, 201 sera samples from 84 RT-PCR confirmed COVID-19 patients with a 10-month follow-up period since symptom onset were included. Each patient’s sample was analyzed on six commercial assays. As previously reported, the serological kinetics showed a high degree of heterogenicity that was patient-dependent but we also reported that these differences were also assay-dependent (Figure 1) [17,18]. The performance of these assays up to 15 days since symptom onset was particularly low because of the natural dynamics of the production of immunoglobulins [2,7,8]. Assays targeting total antibodies presented higher positivity rates and reached their highest positivity rates sooner than IgG assays. The inter-assay agreement was also higher between these total assays. The stratification by disease severity, expressed in this study by the patient’s hospitalization status, showed a higher serological response in severe cases, which is consistent with previous observations [7,8,17,19,20,21]. The Phadia S1 IgG assay had a low performance to detect past SARS-CoV-2 infection compared with other assays. The manufacturer could probably consider redefining the cut-off, as has already been done for other assays, in order to improve the sensitivity [22,23,24,25]. However, in this study, only the cut-offs of the manufacturers were used so that there was no advantage for one method over another. Interestingly, the evaluation of the kinetic models demonstrated that assays targeting total antibodies consistently showed an increase of the antibody titer, at least in hospitalized patients (Figure 1, left panel). The same tendency was also observed in non-hospitalized patients except for the test that targeted antibodies directed against the NCP. On the other hand, assays targeting antibodies directed against the S1 subunit showed a slight decrease in antibody titers except for the DiaSorin S1 + S2 IgG in hospitalized patients (Figure 1, right panel). The drop in cumulative probability of positive samples was consistently highest for the Phadia S1 IgG compared with all other tests (Table 4). The Ortho S1 total antibody assay performed better than the Roche NCP total antibody assay but was not statistically different from the Roche RBD total antibody assay. The Roche RBD and NCP total antibody assays also performed better than all IgG assays. No statistically significant differences were observed between the DiaSorin S1 + S2 IgG and the Ortho S1 IgG (Table 4).

Multiple studies have evaluated the long-term kinetics of anti-SARS CoV-2 antibodies using various assays. A sustained antibody response against the NCP antigen using the Roche NCP total antibody assay was found in several studies, i.e., between three and seven and a half months [20,26,27]. A maintained antibody response against the RBD antigen, as assessed by the Wantai and the Siemens total antibody assays, was also observed up to four months [20,27]. A decrease in anti-RDB IgG and anti-spike IgG levels was similarly observed over a period of up to five months in recent reports [28,29,30] and a significant decrease in sensitivity was also found in studies with up to five months of follow-up with the Abbott assay, which was directed against NCP IgG [20,27,31]. The YHLO assay, which detects both anti-NCP and anti-S IgG, showed high sensitivities from five weeks to three months after symptom onset [32]. Wajnberg et al. found stable antibody titers over a period of at least three months and only modest declines at the five-month time point [33].

In our study, the sustained antibody response as observed with total antibody assays (NCP and RBD) compared with IgG assays may be due to the additional response of non-IgG antibody isotypes. However, the reasons for the differences in assay performance over time for assays targeting the same antigen remain unclear [27]. The nature and structure of the target itself (for example, purified vs. recombinant, full-length vs. truncated, eukaryotic vs. prokaryotic expression system) as well as the protocol definition for determining the cut-off may, at least in part, affect the variability of inter-assays [7].

Whether the antibodies measured with commercial assays have a neutralizing capacity is paramount for indicating the potential level of protective immunity against SARS-CoV-2 infection. Recently, Padoan et al. found that the Ortho S1 IgG (R^2^adj = 0.544) and DiaSorin S1 + S2 IgG (R^2^adj = 0.402) assays were more correlated to neutralization activity compared with the Ortho S1 total antibody (R^2^adj = 0.117) and Roche NCP total antibody (R^2^adj = 0.046) assays [6]. The fact that anti-NCP assays showed a low correlation with the neutralizing capacity was expected as neutralizing antibodies are directed against the spike protein that is responsible for enabling the entry of the virus into the cells that express ACE-2 [22]. A strong correlation between the levels of anti-RBD or anti-spike antibodies and the neutralizing capacity has been found in several reports [11,19,28,33,34]. The neutralizing capacity was found to be maintained from one to five months [19,30,35]. However, although modest declines have been observed at three to five months [29,33], a few studies have pointed out a significant decrease of two to four-fold in neutralizing activity up to three months [21,27,36,37,38].

Data with a longer follow-up, i.e., >6 months, are however still sparse in the literature. Recently, Dan et al. found a slightly decreasing but stable antibody response (anti-S IgG, anti-RBD and anti-NCP using ELISAs) in a population of 188 COVID-19 patients, representing a total of 254 samples, with a maximal follow-up of eight months post-symptom onset. Forty-three samples were collected at > 6 months after the initial infection [18]. Positivity rates at six to eight months were 90% (36/40 samples) for anti-S IgG, 88% (35/40 samples) for anti-RBD IgG and 80% (32/40 samples) for anti-NCP IgG. The positivity rate of patients with positive neutralizing antibodies was 90% (36/40) [18]. In a population of 293 patients, Lau et al. also observed a trend towards lower antibody titers and neutralizing activity after seven months since illness onset but with a positivity rate of almost 100% after 30 days using an anti-RBD IgG ELISA assay [17]. A correlation of 0.53 was found between the ELISA assay and the neutralizing activity. They also found a stronger antibody response in severe patients compared with mildly-infected patients [23]. Ripperger et al. found that anti-RBD, anti-S2 and neutralizing antibodies remained detectable through five to seven months after illness [39].

In a population of 25 COVID-19 patients with a maximal follow-up of eight months, Hartley et al. observed that anti-NCP and anti-RBD IgG were found in each of the 24/25 and 25/25 patients while neutralizing antibodies was detected in 22/25 patients. They noted a decline in neutralization titers and antibody levels with time [14]. Nevertheless, they noted the persistence of SARS-CoV-2-specific B-memory cells, which could represent a more robust surrogate of long-lived humoral immune responses compared with antibodies [15].

It is important to remember that a few patients may develop specific antibodies but may not have detectable neutralizing antibodies. These are only correlation studies that are not related to direct measures of neutralizing activity [27]. The fact that neutralizing antibodies constitute a major protective mechanism against SARS-CoV-2 infection deserves further investigation [17,27,33]. A few differences between various neutralization assays, e.g., pseudo-particle neutralization, microneutralization, fluorescent focus reduction assays, microneutralization assays, plaque reduction neutralization tests, also exist with microneutralization tests found to be less sensitive than plaque reduction neutralization assays [17,40].

The cellular measurements of the immune response have been proposed to be reliable markers for the maintenance of immunity following natural infection or vaccination [14,41,42]. Such approaches should be explored more. Even if previous exposure to SARS-CoV-2, either by true infection or by exposure to a vaccine, significantly decreases the risk of further positive RT-PCR tests [43,44,45,46,47], total immunity might not be guaranteed in all individuals because reinfection with SARS-CoV-2 exists [44,45,46,47,48,49].

Our study has a few limitations. We were not able to perform a neutralization assay at the time. The specificity of each assay was also not determined in this study including the cross-reactivity to common coronaviruses.

## 5. Conclusions

This study shows that assays are not equal for detecting past SARS-CoV-2 infection or investigating seroprevalence in samples for up to 10 months since symptom onset. Assays targeting the total antibody response have the highest positivity rates and perform better than tests targeting only IgG. The waning effect reported in several studies should be interpreted with caution because it may mostly depend on the assay considered. Even if previous exposure to SARS-CoV-2 decreases the risk of subsequent SARS-CoV-2 positivity, total immunity might not be guaranteed in all individuals. Further studies are required to correlate the seropositivity after such a long period post-infection with appropriate serological neutralization assays.

## Figures and Tables

**Figure 1 microorganisms-09-00556-f001:**
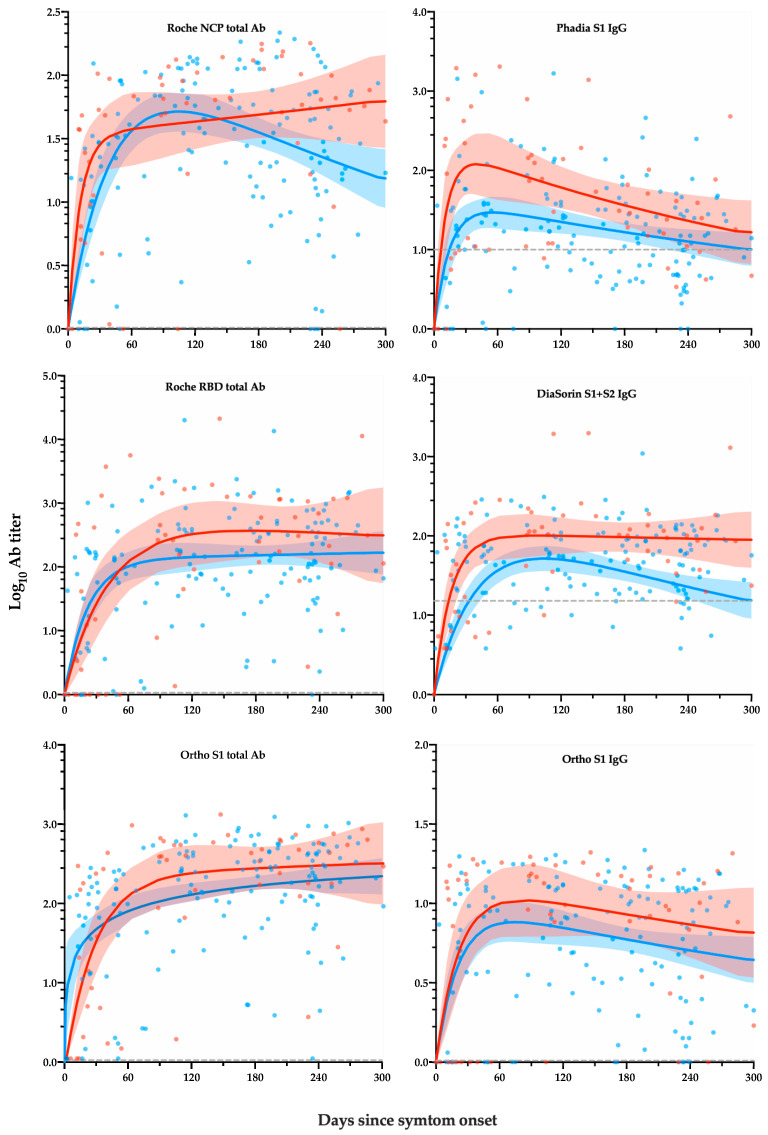
Level of antibody response by days after symptom onset according to severity. Depending on the assay and/or the population considered, a rapid increase in antibody titers was observed followed by a plateau phase or a decrease phase. Blue curves (and 95% CI) and points represent non-hospitalized patients. Red curves (and 95% CI) and points represent hospitalized patients. The dotted grey line corresponds to the manufacturer’s cut-off for positivity.

**Figure 2 microorganisms-09-00556-f002:**
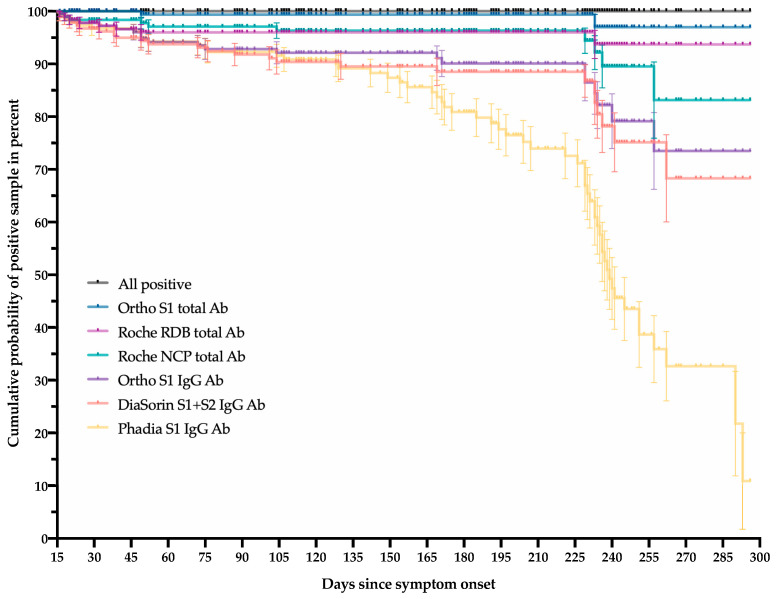
The cumulative probability of positive samples after 14 days until the last follow-up point, i.e., 300 days, using six different commercial assays.

**Table 1 microorganisms-09-00556-t001:**
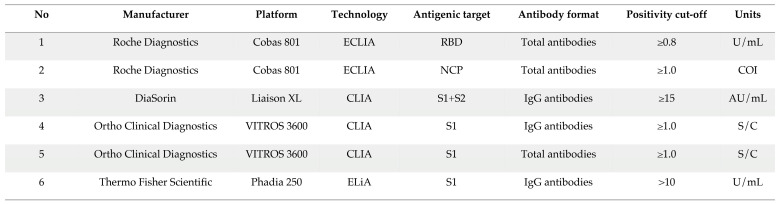
Characteristics of the six assays used in this study.

**Table 2 microorganisms-09-00556-t002:**
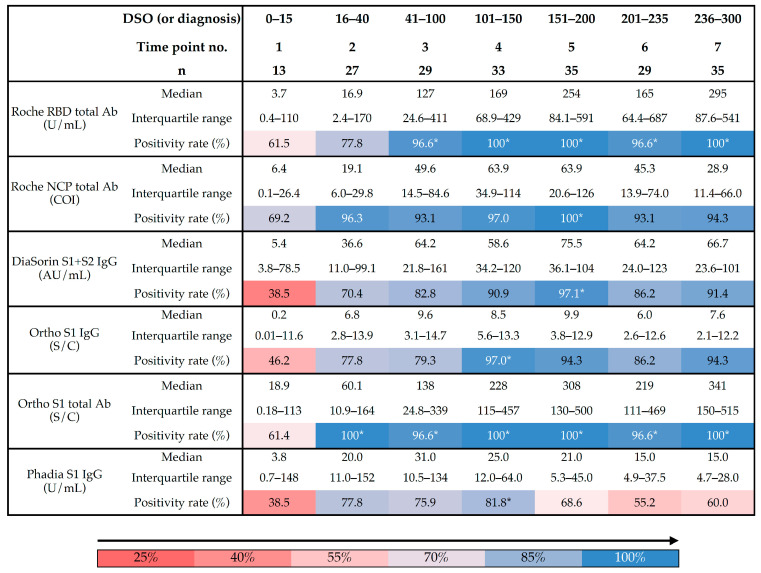
Positivity rates according to different time points using six different assays. * represents maximal positivity rates observed. DSO = days since symptom onset.

**Table 3 microorganisms-09-00556-t003:**
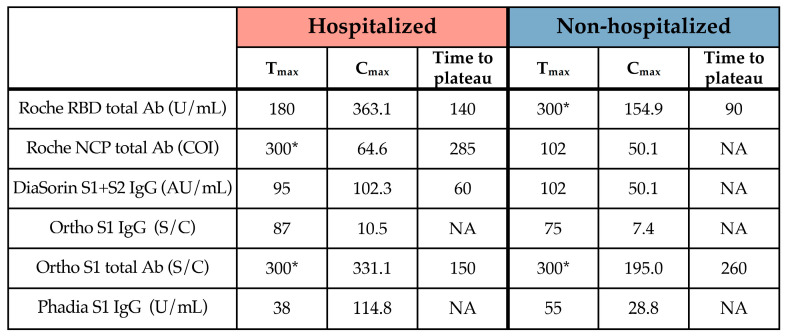
T_max_, C_max_ and time to plateau of the six assays subdivided into hospitalized and non-hospitalized patients. * corresponds to the last time point assessed.

**Table 4 microorganisms-09-00556-t004:**
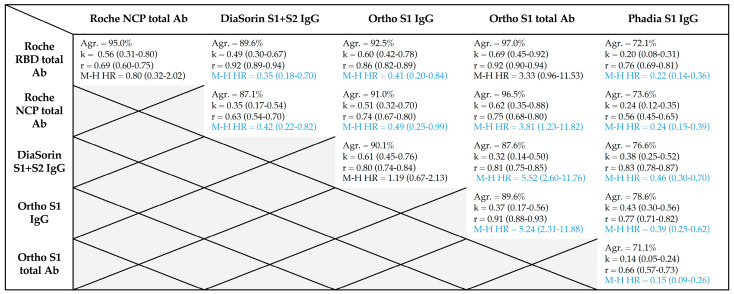
Agreement (Agr.), Cohen’s kappa index (k), correlation (r) and Mantel–Haenszel hazard ratios (M–H HR) between the different assays. Results in blue are statistically significant.

## Data Availability

The data presented in this study are available on request from the corresponding author. The data are not publicly available due to ethical issues.

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
