# Peer review of "Persistence of Anti-SARS-CoV-2 Antibodies Depends on the Analytical Kit: A Report for Up to 10 Months after Infection"

_microorganisms, 2021, doi:10.3390/microorganisms9030556_

Round 1

Reviewer 1 Report

The paper represents a tremendous work on comparing six different analytical kits for SARS-CoV-2 antibody detection.

The aim is stated clearly. Appropriate and key studies are included. The study methods are valid and reliable. There are enough details provided in order to replicate the study. The data is presented in an appropriate way. Results are discussed from different angles and placed into context without being over-interpreted.

The study design is appropriate to answer the aim. The article is consistent within itself.

There are no major flaws of this article.

Specific comments on weaknesses of the article and what could be improved:

Major points  - none

Minor points

  1. Please, state the limitations of the study
  2. Could you please discuss the immunological dynamics of immunoglobulins production? This might reflect the low detected levels between 0-15 days after the onset of the symptoms.
  3. Line 38-39 - "The detection of anti-SARS-CoV-2 antibodies represents an additional method for the 38
    diagnosis of COVID-19" - this has to be precise, since the diagnosis of COVID-19 does not rely on immunoglobulins to SARS-CoV-2 detection.
  4.  The conclusion has to be improved by adding all the concluding remarks.

Author Response

Reviewer:1

The paper represents a tremendous work on comparing six different analytical kits for SARS-CoV-2 antibody detection.

The aim is stated clearly. Appropriate and key studies are included. The study methods are valid and reliable. There are enough details provided in order to replicate the study. The data is presented in an appropriate way. Results are discussed from different angles and placed into context without being over-interpreted.

The study design is appropriate to answer the aim. The article is consistent within itself.

There are no major flaws of this article.

Specific comments on weaknesses of the article and what could be improved:

Major points  - none

  • Thank you very much for your comment!

Minor points

Please, state the limitations of the study

  • A limitation section at the end of the discussion has been added.

Could you please discuss the immunological dynamics of immunoglobulins production? This might reflect the low detected levels between 0-15 days after the onset of the symptoms.

  • Overall, it is not recommended to perform an antibody test before 2 weeks since symptom onset (early antibody kinetics) because of the low level of antibody detected by common assays (Bohn et al. 2020 CCLM, Gillot et al. 2020 JCM, Favresse et al. 2020 CCLM). A sentence has been added in the discussion.

Line 38-39 - "The detection of anti-SARS-CoV-2 antibodies represents an additional method for the diagnosis of COVID-19" - this has to be precise, since the diagnosis of COVID-19 does not rely on immunoglobulins to SARS-CoV-2 detection.

  • Indeed, the antibody test for diagnosis purpose is used in adjunction of a negative molecular tests in patients presenting with suggestive clinical features. A sentence has been added.

 The conclusion has to be improved by adding all the concluding remarks.

  • The conclusion has been adapted.

Reviewer 2 Report

The manuscript describes the persistence of the antibodies against SARS-CoV-2 from up to 10 months. This, however, depend on the type of test. The manuscript is well written and I recommended it publication in the current formate! 

The most important question in the last few months is, how long will the antibodies after symptomatic and asymptomatic SARS-CoV-2 infection be present? The manuscript deliver an answer of the issue. The most essential point is the type of the kits affecting the results outcome. On the top of that, the targeted part of the virus is crucial to estimate the assay sensitivities and specificities. The authors have discussed that the target gene is important. Another significant issue need to be addressed is the cross-reactivities antibody to other coronaviruses. As most of the people have existing antibodies to the “old” coronavirus.

Author Response

Reviewer:2

The manuscript describes the persistence of the antibodies against SARS-CoV-2 from up to 10 months. This, however, depend on the type of test. The manuscript is well written and I recommended it publication in the current formate!

The most important question in the last few months is, how long will the antibodies after symptomatic and asymptomatic SARS-CoV-2 infection be present? The manuscript deliver an answer of the issue. The most essential point is the type of the kits affecting the results outcome. On the top of that, the targeted part of the virus is crucial to estimate the assay sensitivities and specificities. The authors have discussed that the target gene is important. Another significant issue need to be addressed is the cross-reactivities antibody to other coronaviruses. As most of the people have existing antibodies to the “old” coronavirus.

  • Thank you very much for your comment!